# AI-Driven Robust Kidney and Renal Mass Segmentation and Classification on 3D CT Images

**DOI:** 10.3390/bioengineering10010116

**Published:** 2023-01-13

**Authors:** Jingya Liu, Onur Yildirim, Oguz Akin, Yingli Tian

**Affiliations:** 1Department of Electrical Engineering, The City College of New York, New York, NY 10031, USA; 2Memorial Sloan Kettering Cancer Center, New York, NY 10065, USA

**Keywords:** kidney and renal mass segmentation, renal mass classification, weakly supervised learning

## Abstract

Early intervention in kidney cancer helps to improve survival rates. Abdominal computed tomography (CT) is often used to diagnose renal masses. In clinical practice, the manual segmentation and quantification of organs and tumors are expensive and time-consuming. Artificial intelligence (AI) has shown a significant advantage in assisting cancer diagnosis. To reduce the workload of manual segmentation and avoid unnecessary biopsies or surgeries, in this paper, we propose a novel end-to-end AI-driven automatic kidney and renal mass diagnosis framework to identify the abnormal areas of the kidney and diagnose the histological subtypes of renal cell carcinoma (RCC). The proposed framework first segments the kidney and renal mass regions by a 3D deep learning architecture (Res-UNet), followed by a dual-path classification network utilizing local and global features for the subtype prediction of the most common RCCs: clear cell, chromophobe, oncocytoma, papillary, and other RCC subtypes. To improve the robustness of the proposed framework on the dataset collected from various institutions, a weakly supervised learning schema is proposed to leverage the domain gap between various vendors via very few CT slice annotations. Our proposed diagnosis system can accurately segment the kidney and renal mass regions and predict tumor subtypes, outperforming existing methods on the KiTs19 dataset. Furthermore, cross-dataset validation results demonstrate the robustness of datasets collected from different institutions trained via the weakly supervised learning schema.

## 1. Introduction

Kidney cancer is among the ten most common cancers. Early detection often leads to successful treatment and improves survival rates. Medical imaging, such as computer tomography (CT), can effectively reveal the internal structure of the abdomen and is commonly applied for physical examination to detect and diagnose early signs and symptoms of cancer. Finding anatomical information such as the contours, locations, and volumes of kidney tumors plays an essential role in clinical diagnosis. Among kidney masses, renal cell carcinoma (RCC) is the most common renal malignancy, causing many morbidities and mortalities [1,2,3]. The most common RCCs are clear cell, chromophobe, oncocytoma, papillary, and other RCC subtypes. Classifying histologic subtypes of renal cells is crucial to avoid unnecessary biopsy or surgery.

Artificial intelligence has shown the potential to assist clinical care to detect and diagnose abnormal regions in an efficient way [4,5]. State-of-the-art AI algorithms can effectively differentiate benign from malignant renal masses on CT scans [6,7,8]. However, the performance of AI algorithms still needs to be improved when distinguishing different kidney cancer subtypes or making tumor assessments [9]. Additionally, the amount of data in existing studies is limited and lacking in robustness for images collected from various scenarios. Weakly supervised methods have recently been studied for semantic segmentation in natural scene images. According to the weak annotations applied for CNN training [10], these approaches can be divided into four main categories: bounding boxes [11], scribbles [12], points [13], and image-level labels [14]. However, for medical imaging, weakly supervised methods still remain explored.

This paper proposes a novel end-to-end kidney renal mass diagnosis framework consisting of a 3D segmentation network (3D Res-UNet) to automatically segment kidney and renal mass regions and a subtype classification network to accurately identify the subtypes of renal masses using a dual-path approach. In addition, we introduce a simple yet effective weakly supervised training schema to improve the robustness of the proposed framework on cases from different institutions. The main contributions of this paper are summarized as follows:We propose a novel kidney and renal mass diagnosis framework integrating 3D segmentation and renal mass subtype classification. It provides an easy-to-analyze 3D morphologic representation of the kidney and renal mass with the subtypes. The segmentation method adopts the basic 3D U-Net structure with residual blocks included for gaining the cross-layer connections. The postprocessing steps further improved the accuracy and reduced false positives by small region detection. The classification network applies the dual-path schema to combine the different fields of view for the prediction of subtypes.We propose a weakly supervised method to improve the robustness of the trained model on the datasets collected by various vendors with only a few slice-level annotations.The experimental results on the KiTs19 dataset demonstrate the state-of-the-art performance on kidney and renal mass segmentation and classification. Additionally, the results on three NIH datasets (the TCGA-KICH [15], TCGA-KIRP [16], and TCGA-KIRC [17] datasets) show that the proposed framework can be robust in different institutions with few annotations.

The remainder of this manuscript is organized as follows. Section 2 introduces the related work on kidney renal mass segmentation and classification. Section 3 describes the proposed method. The implementation details, experimental results, and discussions are provided in Section 4. Finally, Section 5 summarizes the remarks of this paper.

## 2. Related Work

### 2.1. 3D Semantic Segmentation

3D semantic segmentation in medical imaging refers to the voxel-wise segmentation of abnormalities in 3D imaging (such as CT or MRI). Various applications of semantic segmentation have been proposed, including targeted radiotherapy [18] and patient-specific surgical simulation [19]. U-Net [20] is proposed for semantic segmentation based on a neural network that only performs convolutions (a fully convolutional network (FCN) [21]). The U-Net category is a group of encoder-decoder networks which has been widely used for segmentation tasks with various medical imaging modules and protocols. Many researchers have made various improvements to the U-Net category, residual connections [22], dense connections [23], attention mechanisms [24], squeeze-and-excitation networks [25] or dilated convolutions [26], and V-NET [27].

However, with a standard network architecture, nnU-Net (no new U-Net) [28] is proposed. Extensive experiments show that the original U-Net network is often more robust than its variants. Based on this, the network’s generalization capability can be greatly improved by adjusting hyperparameters, such as batch size, optimizer parameters, patches, and kernel size. nnU-Net performed best in the KiTs19 Challenge. This study shows that in 3D biomedical imaging, dataset properties such as imaging modalities, image sizes, (anisotropic) voxel spacing, and analogies vary widely. The process can be cumbersome, with few transitions from successfully configuring one dataset to another. U-Net was initially proposed in 2D and recently extended to the 3D domain, as in, for example, 3D U-Net [29]. Inspired by nnU-Net and 3D U-Net, we propose a simple 3D U-Net architecture using residual blocks for 3D CT images assisted by hyperparameter tuning during neural network training, data augmentation, and post-processing. We properly process and sample 3D CTs, modify the network architecture, and conduct post-processing on segmented regions.

### 2.2. Weakly Supervised Learning

Semantic segmentation tasks require precise per-pixel annotation, which is very expensive and time-consuming. Recently, weakly supervised methods have been proven efficient for training with low annotation cost and have been shown to improve training efficiency [30]. Usually, weakly supervised methods take four levels of supervision: bounding boxes [31], scribbles [32], points [13], and image-level labels [33]. Many methods focus on weakly supervised deep learning on image-level labels. Wei et al. [34] proposed a simple-to-complex (STC) framework using only image-level annotations to learn deep CNN (DCNN) models for semantic segmentation. The framework first trains an initial DCNN on the clean images (only an object and background) and then trains Enhanced-DCNN with the segmentation masks predicted from Initial-DCNN and image-level annotations. Finally, a Powerful-DCNN for semantic segmentation is prepared based on the Enhanced-DCNN and image-level annotations. To further enrich the information on image-level annotation, Hisham et al. [35] proposed an image-level supervised method to construct an object category density map by catching both the global object count and the spatial distribution of object instances, which has been proven effective in image-level supervised instance segmentation. There are few weakly supervised methods explored in the medical imaging field. Recently, Sadeghi [36] proposed a weakly supervised semantic segmentation approach based on image-level labels for kidney tumor segmentation. Image-level annotation is referred to as the image-level label, such as “0” representing no lesion and “1” representing a lesion. However, the image-level label only provides high-level annotations. Without accurate annotations, it may lead to false-positive predictions. This paper requires image-level segmentation annotation but only with very few slices.

### 2.3. Kidney and Renal Mass Segmentation and Classification

Deep learning-based algorithms have been applied to renal mass segmentation and classification [37,38,39,40,41]. Yang et al. [42] proposed a three-dimensional (3D) fully convolutional network (FCN) approach with a pyramid pooling module (PPM) to segment kidney and tumor regions in CT images. The proposed network was applied to 3D volumetric images with rich 3D spatial contextual information for accurate kidney and tumor segmentation compared with 2D networks. Yu et al. [43] proposed crossbar patches of the tumor regions (vertical and horizontal patches) to capture global and local appearance information. Two sub-models were trained in a cascaded training manner. Yin et al. [44] developed two CNNs to segment the kidneys automatically. A regression algorithm was applied to the first CNN to predict a 3D bounding box around each kidney, and then a 3D U-Net algorithm predicted the outline of the kidney regions based on the input of the bounding cube.

Recently, Xia et al. [45] proposed a two-stage framework consisting of image retrieval and semantic segmentation. A deep convolutional neural network structure using sparse CNN and ResNet was applied to extract image features to leverage the abdominal CT scan image acquired under various angles. Then, a scale-invariant feature transform flow transform adopts a Markov random field to fuse label information to smooth the prediction of pixels. Ruan et al. [46] proposed a framework composed of a multi-scale feature extractor (MSFE) to target the detection small nodules, a locator of the area of interest (LROI) to reduce the time complexity, and a feature-sharing generative adversarial network (FSGAN) for joint learning and adversarial learning. Many classification tasks were explored, such as classification between benign and malignant masses [37,38,39] and RCC classes (ccRCC, pRCC, and chRCC) [47].

Despite the various studies on the effect of the above methods on the segmentation of renal tumors, few efforts have been made to propose a kidney and renal mass diagnosis framework to accurately quantify the volume of kidney and renal mass as well as predict the subtype of renal masses. Recently, Uhm et al. [48] proposed an end-to-end framework for kidney and renal mass segmentation. However, multi-phase abdominal CTs were applied, which required extra images at multiple contrast-enhanced stages. Chen et al. [49] proposed a weakly supervised method for medical image segmentation using a category-causality chain. The proposed method works to find the boundary of organs but is unsuitable for small-sized nodule segmentation.

## 3. Our Method

We propose a novel kidney and renal mass diagnosis framework, including kidney and renal mass segmentation and subtype of kidney mass prediction. First, the proposed framework extracts the pixel-level kidney and renal mass regions from an entire CT volume via a three-dimensional (3D) segmentation network trained in a fully supervised manner. Then, a dual-path CNN-based classification model predicts subtypes of renal masses. Pathological labels of renal subtypes are used to train the classification model. Further, a weakly supervised method is applied to use annotations of only a few CT slices to improve the robustness of the diagnosis framework on datasets captured from different institutions.

### 3.1. Kidney and Renal Mass Segmentation

We employed 3D U-Net, an encoder-decoder network commonly utilized in image segmentation. Similar to the U-Net [29] framework and nnU-Net framework [28], the encoder consists of five convolutional neural network blocks, which aggregate semantic features but lose spatial information. As spatial information is essential for image reconstruction, a skip connection was introduced in the U-Net decoder. The high-resolution features were obtained from the encoder and mapped to the decoder for the feature concatenation. The residual blocks were applied to the U-Net network architecture to overcome the vanishing and exploding gradient problem.

To enrich the spatial feature representation, we employed 3D Res-UNet to directly predict the 3D volume of the kidney and renal mass from a whole 3D CT volume. The model was trained from scratch and evaluated using 5-fold cross-validation on the training set. We applied a leaked rectified linear unit (ReLU) and instance normalization to optimize the model. The losses were combined with dice loss and cross-entropy loss for a total of 300 training epochs. The network structure is shown in Figure 1. The width of the first convolution block was set to 32 channels, doubling the size in the next convolution block, and we continued downsampling the feature map until the output feature map size reached 4×4×4. At the last layer of the decoder, we generated the kidney and renal mass prediction masks via a convolution layer and then applied SoftMax for output prediction. The skip connection layer enables the network to correctly make predictions at low resolutions and avoid passing an inaccurate feature prediction to the high-resolution layer.

### 3.2. Renal Mass Classification

Unlike previous work, we designed and evaluated a deep learning framework that classifies renal masses into the most common RCCs subtypes: clear cell, chromophobe, oncocytoma, papillary, and other RCC subtypes, using abdominal CT scans as input data. The proposed framework segments kidney and renal masses, detects renal mass categories in a unified framework and diagnoses only through CT data without manual intervention, which can effectively assist clinical medical detection.

We first automatically segmented the regions of kidney and renal masses at the pixel level via the 3D segmentation network. Once the kidney and renal mass regions were detected, both the regions of the renal mass with the kidney and only the renal mass were cropped from the CT images. The cropped region helped focus on only the relevant information for subtype classification. The cropped kidney and renal mass CT images were further employed as the input to the renal mass subtype classification network.

A dual-path learning schema was applied to enrich feature learning, which can help the network simultaneously learn the correlation between the local and global feature representations. It is essential to preserve the global structures around the renal mass region and texture details in the renal mass region. The global path took the input of the region with both kidney and renal mass, and the local path took the input of only the renal mass region. We applied ResNet-50 [22] as the backbone network for the classification training. The global and local networks applied the same network structure, comprised of five convolution layers, each with a skip connection, ReLU activation, and a stride of 2. The five layers were comprised of (64,128,256,256,512) channels of three-by-three filters. The output of the fifth layer was flattened and transformed into a vector of size 64 through a fully connected layer. The global and local network features were concatenated and followed by a six-class soft-max classifier for subtype prediction. A dropout of 50% was implemented in the fully connected layer. For the fourth and fifth layers, we applied the dense convolutional layer that obtains additional inputs from all preceding layers and passes on its feature maps to all subsequent layers by concatenation. Each layer received features that collect the information from all preceding layers.

### 3.3. Weakly Supervised Learning

To improve the robustness of the segmentation network, we applied a weakly supervised method that only requires a small number of pixel-level annotated CT slices based on a trained network. Annotations of only three slices from each volume were needed. The three slices were chosen based on empirical experiments. For qualitative assessment, we trained the network on three annotation samples: (1) the first slice, (2) the slice with the largest diameter of the renal mass, and (3) the last slice of the renal mass. The network can predict the complete 3D segmentation from only a few labeled slices, saving the cost of full annotation.

To assess quantitative performance in semi-automated settings, we cross-validated all annotation slices in all different samples from the training data with five-fold cross-validation with and without batch normalization. The intersection of IoU is used as a precision measure to compare the predicted 3D volumes.

## 4. Experimental Results and Discussions

### 4.1. Datasets

The public dataset KiTs19 [50] was applied to train the segmentation model. The KiTs19 dataset collects arterial CT images of patients undergoing nephrectomy for a renal mass in a medical center. A total of 210 3D CTs from 210 patients were selected from the publicly published 2019 MICCAI Renal Tumor Segmentation Challenge training set for model training and testing. Among all the scans, 203 (67.7%) of clear cell RCC, 28 (9.3%) were of papillary RCC, 27 (9%) were of chromophobe RCC, 16 (5.3%) were of oncocytoma, and 26 (8.7%) were of other types. For testing, we employed three additional publicly available NIH datasets for performance evaluation across different institutes: TCGA-KICH [15] (15 cases), TCGA-KIRP [16] (20 cases), and TCGA-KIRC [17] datasets (40 cases). These NIH datasets contain cases with various tumor shapes and tumor sizes (on average 23% larger) compared with the KiTs19 dataset.

### 4.2. Experimental Settings

#### 4.2.1. Pre-Processing

CT images were converted from the Neuroimaging Information Technology Program (Nifti) format to DICOM. Following nnU-Net, CT scans were resampled by the standard spacing of 3.22×1.62×1.62 mm to resize the CT images to 128×248×248 voxels. A Hounsfield unit (HU) represents the linear transformation of the measured attenuation coefficient from −1000 (air) to 2000 (bone). To better analyze the kidney areas, the window width and level were set to 400 and 30, respectively. Then, the pixel values were normalized in the range of 0 to 1. Image augmentation was implemented during the training, including random rotations ((−45,55) degrees with a probability of 0.2), random scaling (scaling factor of 0.9 and 1.1 with a probability of 0.2), random elastic deformations (with a probability of 0.05), gamma correction augmentation (with a nonlinear intensity transformation of 0.7 and 1.5 per pixel with a probability of 0.15), and mirroring (with a probability of 0.5).

#### 4.2.2. 3D Semantic Segmentation

**Training.** The patch size was set to 128×128×128, and the batch size was set to 2 with one NVIDIA GTX 1080ti. The Adam stochastic optimizer was implemented with a beta of 0.99. The initial learning rate was set to $5\times 10^{-3}$ to learn the network weights and decayed based on the “poly” learning rate policy, ${\left(1-epoch/epoch_{max}\right)}^{0.9}$. Grid search was applied to choose the best learning rate from ($1\times 10^{-3},3\times 10^{-3},5\times 10^{-3},1\times 10^{-4},5\times 10^{-4}$). The sum of cross-entropy and dice losses was applied to train and evaluate the model to improve the accuracy and training stability. The model was trained with 300 epochs.

**Post-processing.** Connected component-based post-processing has effectively eliminated falsely detected regions on medical image segmentation [28,51]. The algorithm suppresses false predictions by removing a small discontinuous predicted region and keeping the most relevant reference component. We followed this algorithm to remove all but the top 2 largest components and automatically suppress the small and discontinuous components, which were found to be beneficial for the combined kidney and renal mass region [28]. For inference, the post-processing step was performed for all testing samples. During the training, if removing a region could improve the loss of the dice coefficient, it was considered a false positive and removed from the component. This step helps the network focus on punishing the hard false-positive samples.

#### 4.2.3. Renal Mass Classification

We first cropped kidney and renal mass regions from the detected CT slices. The cropped images were resized to 64×64 as the input of the classification network. The classification model was trained with 100 epochs and optimized by the Adam optimizer with a beta set to 0.99. We applied the weighted cross-entropy loss function. The batch size was set to 32 with an initial learning rate of 0.001. The learning rate was reduced by 0.1 for every 25 epochs.

### 4.3. Evaluation Metrics

**Segmentation.** The kidney and renal mass were evaluated by the following metrics. The Dice similarity coefficient was applied as an evaluation metric to calculate the spatial overlap index between the predicted 3D volume heatmap and the radiologist’s estimation. It estimates the accuracy of the structure’s volume and is sensitive to heatmaps with different shapes but similar volumes. The Dice value ranges from zero for no spatial overlap to 1 for complete overlap. In addition, we applied pixel accuracy, sensitivity, and specificity to evaluate the model performance. Pixel accuracy was calculated via the ratio of correctly classified pixels to the total number of pixels. The specificity refers to the ratios of pixels that were correctly classified to the number of pixels to all masks. The sensitivity represents the ratios of negative pixels correctly classified as negative to all negative ones.

**Classification.** The area under the curve (AUC) was applied to evaluate the classification performance.The AUC is calculated by multiclass classification via the one-vs.-rest scheme with 95 confidence intervals. For each step, a class is considered a positive class, and others are negative classes to calculate the ROC to obtain the AUC scores. A permutation test compared the model’s performance (sensitivity and specificity). Five-fold cross-validation was conducted for the results of each dataset.

### 4.4. Experimental Results and Analysis

#### 4.4.1. Comparison with the State-of-the-Art Methods

Our model predicts remarkably accurate kidney segmentation on the KiTs19 testing set, with an average Dice metric of 96.7% and 86.6% for kidney and renal mass regions, respectively. The proposed framework takes advantage of the rich spatial information provided by 3D Res-UNet and reaches state-of-the-art performance on kidney and renal mass segmentation. Our method has the best segmentation performance compared to the state-of-art deep learning methods for segmenting renal lesions on the KiTs19 dataset, as shown in Table 1. The pixel accuracy is 95.3%, the Dice coefficient is 86.6%, the sensitivity is 91.2%, and the specificity is 88.1%. Compared to the existing work [43,44,45,46], our method takes advantage of applying the residual blocks and extensive hyperparameter tuning with the appropriate pre-processing and post-processing steps. Better segmentation results are thus obtained.

#### 4.4.2. Model Robustness

Additionally, we conducted experiments to assess the robustness of the proposed method on the TCGA-KICH, TCGA-KIRP, and TCGA-KIRC datasets. These three testing datasets have various tumor shapes and different tumor sizes (on average 23% larger) compared to the training samples, especially the TCGA-KIRC dataset. As deep learning methods are data-driven, the Dice coefficient scores of cross-validations are slightly decreased. We observe that the inaccurately predicted regions are on the edges of very large tumors, barely seen in the training samples. The weakly supervised learning method on these datasets further improves the model performance by carrying out fine-tuning with few slices. With the testing dataset being distinct from the training samples, the Dice result is still comparable, demonstrating the robustness of the proposed deep learning method.

**TCGA-KICH** (cases with chromophobe subtypes). With the weakly supervised method, the Dice accuracy of the kidney region is 91.2%, and the Dice accuracy renal mass region is 83.4%, which represents improvements of 5.3% and 4.2% compared to without the weakly supervised method. The subtype prediction is 84.5%, representing a 2% improvement with the weakly supervised method.

**TCGA-KIRP** (cases with papillary subtypes). With the weakly supervised method, the Dice accuracy of the kidney region is 92.8%, and the Dice accuracy of the renal mass region is 81.8%, which represents improvements of 3.3% and 5.4% compared to without the weakly supervised method. The subtype prediction is 79.5%, representing a 2% improvement with the weakly supervised method.

**TCGA-KIRC** (cases with clear cell subtypes). With the weakly supervised method, the Dice accuracy of the kidney region is 93.2%, and the Dice accuracy of the renal mass region is 80.1%, which represents improvements of 2.5% and 1.8% compared to without the weakly supervised method. The subtype prediction is 86.1%, representing a 2.5% improvement with the weakly supervised method.

The visualization of the kidney region (red) and renal mass region (blue) overlaps with the original CT images shown in Figure 2. The visualization demonstrates the effectiveness of the proposed method in accurately segmenting the kidney and renal mass regions.

#### 4.4.3. Renal Mass Subtype Classification

Table 2 shows the areas under the curves (AUCs) of the model with a 95% confidence interval (CI) for the ROC curves. The model achieved an average AUC of 85% for all subtype classes. We observe that the model correctly classifies the clear cell, papillary, and chromophobe subtypes with AUCs of 87.1%, 86.2%, and 85.9%, respectively. The model achieves an accuracy of 79.9% for oncocytoma, as this subtype is often difficult to differentiate from RCC because of their similarities in morphologic, histologic, and imaging characteristics. The proposed model successfully classified fine-grained renal masses into five major pathological subtypes.

### 4.5. Ablation Study

We investigated the effectiveness of the proposed semantic segmentation framework with an ablation study, in which 3D U-Net was applied as a baseline model. Here, 3D Res-UNet uses the 3D U-Net network structure and involves residual connections to all convolutional blocks. 3D Res-UNet with preprocessing and post-processing are implemented individually, as shown in Table 3. Our emerging modifications of 3D U-Net effectively improve the accuracy of the network compared to the baseline 3D U-Net segmentation network.

## 5. Conclusions

This paper has proposed an end-to-end kidney and renal mass diagnosis framework that can quantify renal mass morphology details to monitor disease progression. The proposed weakly supervised method improves the generalization ability of the network. The strategy of dual-path training of subtype classification not only improves the prediction accuracy but also prevents the false-positive prediction of renal masses. The experimental results demonstrate the effectiveness of the diagnosis framework and its potential to be a clinical tool.

## Figures and Tables

**Figure 1 bioengineering-10-00116-f001:**
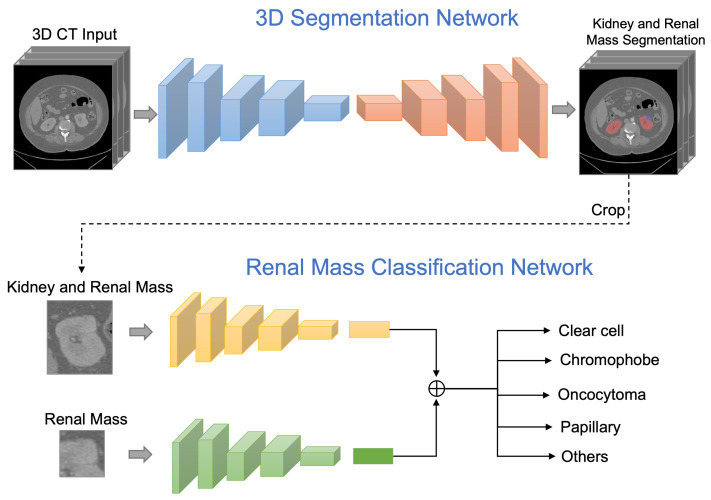
The proposed framework first predicts the kidney and renal mass regions via a 3D segmentation network. The classification network then takes the inputs of the regions of renal mass and renal mass with kidney, then outputs a probability distribution over subtype classes of the renal mass.

**Figure 2 bioengineering-10-00116-f002:**
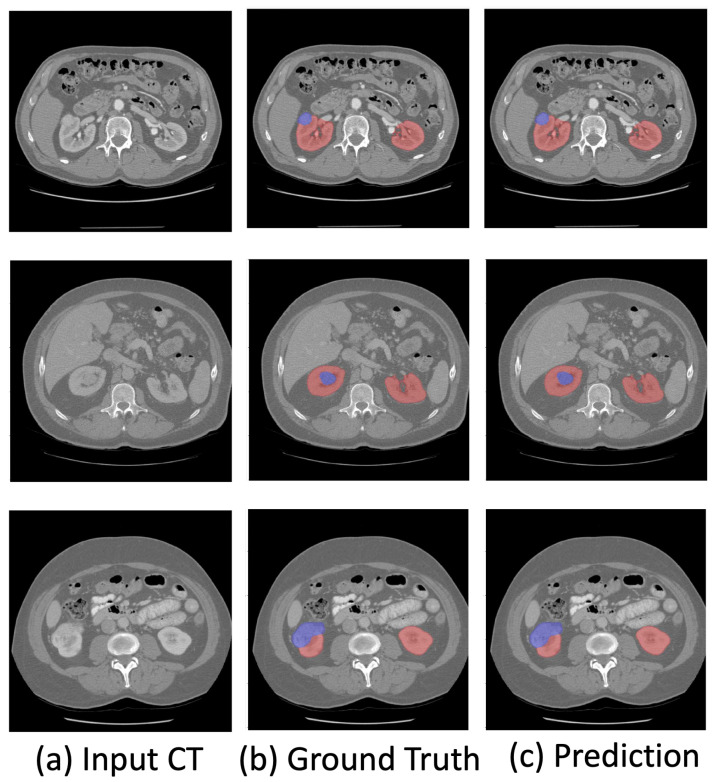
Visualization of the predicted kidney (red) and renal mass (blue) segmentation.

**Table 1 bioengineering-10-00116-t001:** Quantification results of our proposed method and the state-of-art renal mass segmentation methods on the KiTs19 dataset.

Methods	Pixel Accuracy	Dice Coefficient	Specificity	Sensitivity
Xia et al. [45]	92.1%	79.6%	86.7%	83.4%
Yin et al. [44]	89.4%	83.8%	81.1%	85.4%
Yu et al. [43]	87.7%	80.4%	83.4%	82.2%
Ruan et al. [46]	95.7%	85.9%	89.4%	86.2%
**Ours**	**95.9%**	**86.6%**	**91.2%**	**88.1%**

**Table 2 bioengineering-10-00116-t002:** Quantification results of renal mass subtype classification on KiTs19 datasets.

Subtype	AUC	Specificity	Sensitivity
Clear Cell	87.1%	84.3 %	84.1%
Papillary	86.2%	83.7%	82.4 %
Chromophobe	85.9%	82.1%	83.7 %
Oncocytoma	79.9%	77.7%	76.9 %
Others	85.8%	81.6%	80.4 %

**Table 3 bioengineering-10-00116-t003:** Ablation study of the effectiveness of the proposed method by comparing the performance on the KiTs19 dataset: Dice coefficients and sensitivity.

3D U-Net	ResNet	Preprocessing	Post-Processing	Pixel Accuracy	Dice Coefficient	Specificity	Sensitivity
*√*	-	-	-	86.2%	80.9%	84.1%	78.8%
*√*	*√*	-	-	89.1%	82.4%	86.8%	83.2%
*√*	*√*	*√*	-	91.7%	83.1%	89.2%	86.4%
*√*	-	*√*	*√*	91.4%	82.7%	87.3%	83.7%
*√*	*√*	*√*	*√*	95.9%	86.6%	91.2%	88.1%

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
