# Peer review of "AI-Driven Robust Kidney and Renal Mass Segmentation and Classification on 3D CT Images"

_bioengineering, 2023, doi:10.3390/bioengineering10010116_

Round 1

Reviewer 1 Report

The manuscript proposes a UNET-based method for kidney and renal mass segmentation and a CNN with two branches for renal cell carcinoma (RCC) subtype classification.
The focus of the study is not clear. The segmentation and the classification tasks are handled by two different neural networks. These two tasks are handled separately in the manuscript. In the meanwhile, both networks need to be introduced in more details. Why not study them in separate reports?

Neither network seems to be innovative. Regular structures and modules are used in the design. If the authors believe the proposed two networks are advantageous, the reasons need to be explained clearly.

For the segmentation network, how the performance indices, including the accuracy, sensitivity and specificity, are calculated? For the multi-class classification network, how is the AUC calculated? Are the sensitivity and specificity defined in the same ways as those in the segmentation task?

When the authors compare the proposed methods with other methods, why if the post-processing is removed? In the post-processing step, when a new case is input, how false positive regions can be determined if no ground truth is available?

It might be a good idea that the study can be split into two separate studies to deal with segmentation and classification.

Author Response

We would like to appreciate the reviewer's constructive comments and insightful suggestions to improve the quality of the manuscript. We have addressed all the raised issues in our point-by-point responses in the attachment and revised the paper accordingly (highlighted in blue font in the revised manuscript).

Reviewer 2 Report

Dear proposed the method based on the CNN model to work on 3D CT images of kidneys to diagnose the disease. I have two significant concerns. 

How can the author diagnose the kidney and renal mass it shows that the author lacks the biological concept of kidney and renal masses of it, and it is necessary to make the proper analysis of images and what features should author required for segmenting to make the proper classification. It needs proper related work, and it is missing in the paper. 

The second primary concern about the results is that because the author compares the results of different researchers targeting different features, the comparison tables of segmentation and classification are required concerning their features or region of interest. 

Author Response

(The authors gave the same response as above.)

Reviewer 3 Report

In section3, a novel kidney and renal mass diagnosis framework, including kidney and renal mass segmentation and subtype of kidney mass prediction, is proposed. Authors should explain the advantages of the framework that includes both of these processes, unlike a framework that treats them separately as in previous works. Is it simply the inclusion of the two processes?

In 3.3, the authors should clearly explain whether the three slices used for annotation are part of the test dataset. And, how to select the slice with the largest diameter should be explained.

In 4.5, there is a lack of explanation of how the specificity is calculated. Is it accuracy for all voxels except for the region of interest? The authors should add an explanation and clarify.

In 4.6.1, is the hyperparameters tuning done in the process including pre-processing and post-processing? It is not explained what the tuned parameters are and how they were tuned. These are important parts of the proposal and needs to be explained clearly.

In 4.6.2, a reviewer doesn't understand what the numbers explained refer to. It should be explained more clearly. For example, it would be easier to understand if a comparison of the accuracy before and after applying fine-tuning is shown in a table.

Confidence intervals for ROCAUC are not shown in Table 2.

The ablation study described in 4.7 shows a comparison of three processes added sequency to the 3D U-Net. Other combinations should also be shown. For example, how about adding only pre-processing and post-processing? In Table 3, only dice coefficients and sensitivity are shown, but specificity and pixel accuracy are not shown. Specificity and Pixel accuracy should be shown as in Table 1.

Author Response

(The authors gave the same response as above.)

Round 2

Reviewer 1 Report

As pointed out previously, a single paper cannot really represent both the segmentation and classification tasks in a fashion that other researchers can reproduce the results solely based on the manuscript. The manuscript cannot be evaluated without those details.

No innovations cannot be clearly identified in the neural networks. This cannot be changed by rephasing the claims in the summary. This problem is there if the experimental design is not re-constructed.

The definitions of sensitivity and specificity are not exactly in accordance with the standard definitions.

Using a standard library does not mean that the usage of a specific performance index is automatically correct. How do the authors explain the AUC value calculated by the scikit-learn?

The problem of the post-processing is due to below step,

"If removing a region can improve the Dice coefficient, it is considered a false positive and removed from the component."

For a new case that the Dice coefficient is unknown, how this step can be implemented?

A lot of experiment details are still missing. The authors added that "Grid search is applied 253 for the hyper-parameter tuning." But how the grid search was performed is still not clear. For example, how many hyperparameters were there? How was the grid designed? How was the best hyperparameter combination picked? This is just one example. The authors should understand that a journal paper should give all the details that one can reproduce their results solely based on their writing.

Author Response

(The authors gave the same response as above.)

Reviewer 2 Report

The authors responded to all comments, and the paper is ready to publish. 

Author Response

We would like to appreciate the reviewer's constructive comments and insightful suggestions to improve the quality of the manuscript.